# On the Identifiability of Nonlinear ICA with Unconditional Priors

**Yujia Zheng**[1]    **Ignavier Ng**[1]    **Kun Zhang**[1,2]
[1] Carnegie Mellon University
[2] Mohamed bin Zayed University of Artificial Intelligence
`{yujiazh,ignavierng,kunz1}@cmu.edu`

## Abstract

Nonlinear independent component analysis (ICA) aims to recover the underlying marginally independent latent sources from their observable nonlinear mixtures. The identifiability of nonlinear ICA is a major unsolved problem in unsupervised learning. Recent breakthroughs reformulate the standard marginal independence assumption of sources as conditional independence given some auxiliary variables (e.g., class labels) as weak supervision or inductive bias. However, the modified setting might not be applicable in many scenarios that do not have auxiliary variables. We explore an alternative path and consider only assumptions on the mixing process, such as *independent influences*. We show under these assumptions that the marginally independent latent sources can be identified from the nonlinear mixtures up to a component-wise (linear) transformation and a permutation, thus providing an identifiability result of nonlinear ICA without auxiliary variables. We provide an estimation method and validate the theoretical results experimentally.

## 1 Introduction

Nonlinear independent component analysis (ICA) is a fundamental problem in unsupervised representation learning. It generalizes linear ICA (Hyvärinen et al., 2001; Comon, 1994) to identify latent sources from observations, which are assumed to be an invertible nonlinear mixture of the sources. For an observed vector $\mathbf{x}$, nonlinear ICA expresses it as $\mathbf{x} = \mathbf{f}(\mathbf{s})$, where $\mathbf{f}$ is an unknown invertible mixing function, and $\mathbf{s}$ is a latent random vector representing the marginally independent sources. The goal is to recover function $\mathbf{f}$ as well as sources $\mathbf{s}$ from the observed mixture $\mathbf{x}$ up to certain indeterminacies. While nonlinear ICA is of general interest in a variety of tasks for decades, such as disentanglement (Lachapelle et al., 2022) and unsupervised learning (Oja, 2002), its identifiability remains a major obstacle for providing a principled framework. The key difficulty is that, without additional assumptions, there exist infinite ways to transform the observations into independent components while still mixed w.r.t. the sources (Hyvärinen & Pajunen, 1999).

To deal with this challenge, existing works introduce the auxiliary variable $\mathbf{u}$ (e.g., class label, domain index, time dependency) and assume that sources are conditionally independent given $\mathbf{u}$ (Hyvärinen & Morioka, 2017; Hyvärinen et al., 2019; Khemakhem et al., 2020). Most of them require the auxiliary variable to be observable, while clustering-based methods (Hyvärinen & Morioka, 2016) and those for time series (Hälvä & Hyvärinen, 2020; Hälvä et al., 2021) are exceptions. These works impose mild restrictions on the mixing process but require many distinct values of $\mathbf{u}$ (e.g., number of classes) for identifiability, which might restrict their practical applicability. Motivated by this, Yang et al. (2022) reduce the number of required distinct values by strengthening the functional assumptions on the mixing process. As described, all these results are based on conditional independence, instead of the standard marginal independence assumption, of the sources to achieve identifiability in specific tasks, i.e., those with auxiliary variables as weak supervision. The identifiability of the original setting focusing on marginal independence, which is crucial for unsupervised learning, stays unsolved.

Instead of introducing auxiliary variables, there exists the possibility to rely on further restrictions on the mixing functions to achieve identifiability, but limited results are available in the literature over the past two decades. With the assumption of conformal transformation, Hyvärinen & Pajunen (1999) proved identifiability of the nonlinear ICA solution up to rotational indeterminacy for only two

sources. Another result is shown for the mixture that consists of component-wise nonlinearities added to a linear mixture (Taleb & Jutten, 1999). MND (Zhang & Chan, 2008) improves the performance of separation by preferring solutions with high linearity, while no identifiability result is given.

In this work, we aim to show the identifiability of nonlinear ICA with unconditional priors under specific conditions. We show that without any kind of auxiliary variables, the latent sources can be identified from nonlinear mixtures up to a component-wise invertible transformation and a permutation under assumptions on the mixing process. Specifically, we study the following conditions on the mixing process: (1) the influence of each source on the observed variables is independent of each other in a non-statistical sense, and (2) factorial Jacobian determinant (e.g., volume-preserving transformation). Different from existing works relying on weak supervision of auxiliary variables (observable or not), we establish an identifiability result in a fully unsupervised setting, which is within the original setting of ICA that is based on marginal independence of the latent sources.

In particular, our work is consistent with previous results on ruling out specific spurious solutions (Gresele et al., 2021) and reducing the number of required values of auxiliary variable (Yang et al., 2022) by restricting mixing function $f$. Besides, we show that the component-wise indeterminacy can be further reduced to the component-wise linear transformations if the true sources are assumed to be Gaussian. This work opens up the possibility of fully identifiable nonlinear ICA under the standard marginal independence assumption. We validate our theoretical claims experimentally.

## 2 NONLINEAR ICA WITH UNCONDITIONAL PRIORS

In this section, we present two identifiability results for nonlinear ICA with unconditional priors. Instead of introducing any type of auxiliary variables or side information, we consider conditions only on the mixing process, which are rather natural in specific real-world scenarios.

### 2.1 IDENTIFIABILITY OF NONLINEAR ICA

We consider the following data-generating process of nonlinear ICA

$$p_{\mathbf{s}}(\mathbf{s}) = \prod_{i=1}^{n} p_{s_i}(s_i), \tag{1}$$

$$\mathbf{x} = \mathbf{f}(\mathbf{s}), \tag{2}$$

where $n$ denotes the number of latent sources, $\mathbf{x} = (x_1, \ldots, x_n)$ denotes the observed random vector, $\mathbf{s} = (s_1, \ldots, s_n)$ is the latent random vector representing the marginally independent sources, $p_{s_i}$ is the marginal probability density function (pdf) of the $i$-th source $s_i$, $p_{\mathbf{s}}$ is the joint pdf, and $\mathbf{f} : \mathbf{s} \to \mathbf{x}$ denotes a nonlinear mixing function. Different from previous works for identifiable nonlinear ICA, sources $\mathbf{s}$ do not need to be mutually conditionally independent given an auxiliary variable. Instead, our setting is consistent with the traditional ICA problem based on a general marginal independence assumption of sources. Besides, no particular form of the source distribution is assumed at this point, which is similar to (Yang et al., 2022) and different from the works that directly assume an exponential family (Hyvärinen et al., 2019; Hälvä & Hyvärinen, 2020; Lachapelle et al., 2022).

Let $\mathbf{g} : \mathbf{x} \to \hat{\mathbf{s}}$ be the estimated unmixing function with $\hat{\mathbf{s}} = (\hat{s}_1, \ldots, \hat{s}_n)$ being the marginally independent estimated sources. Denoting by $\mathbf{J_g}$ the Jacobian matrix of $\mathbf{g}$, we set its columns to be orthogonal to each other and have equivalent euclidean norm, which can be written as $\mathbf{J_g}(\mathbf{x}) = \mathbf{O}(\mathbf{x})\lambda(\mathbf{x})$, where $\mathbf{O}(\mathbf{x})$ is an orthogonal matrix and $\lambda(\mathbf{x})$ is a scalar. This is consistent with the pairwise uncorrelatedness between columns of the Jacobian after centering. We assume that the distribution of the estimated sources $\hat{\mathbf{s}}$ follows a factorial multivariate Gaussian:

$$p_{\hat{\mathbf{s}}}(\hat{\mathbf{s}}) = \prod_{i=1}^{n} \frac{1}{Z_i} \exp\left(-\theta_{i,1}\hat{s}_i - \theta_{i,2}\hat{s}_i^2\right), \tag{3}$$

where $Z_i > 0$ is a constant. The sufficient statistics $\theta_{i,1} = -\frac{\mu_i}{\sigma_i^2}$ and $\theta_{i,2} = \frac{1}{2\sigma_i^2}$ are assumed to be linearly independent. We set the variances $\sigma_i^2$ to be distinct without loss of generality, which can be placed as a constraint during the estimation process as the sufficient statistics of the estimated distribution are trainable. It is noteworthy that regularization on the estimating process is not assumption on the true generating process. The main result is as follows:

**Proposition 2.1.** *Consider the model defined in Eqs. 1-3. Assume*

    *i.* *(Independent Influences): The influence of each source on the observed variables is independent of each other, i.e., each partial derivative $\partial \mathbf{f} / \partial s_i$ is independent of the other sources and their influences in a non-statistical sense.*

    *ii.* *The mixing function $\mathbf{f}$ and its inverse are twice differentiable.*

    *iii.* *The Jacobian determinant of mixing function can be factorized as $\det(\mathbf{J_f(s)}) = \prod_{i=1}^{n} y_i(s_i)$, where $y_i$ is a function that depends only on $s_i$. Note that volume-preserving transformation is a special case when $y_i(s_i) = 1, i = 1, \ldots, n$.*

*Then function $\mathbf{h} := \mathbf{g} \circ \mathbf{f}$ is a composition of a component-wise invertible transformation and a permutation.*

The proof is provided in Appendix A.1. Note that some part of the proof technique is inspired by or based on Yang et al. (2022). To build intuition, one can consider a filmmaking process in a safari park, of which the main characters are wild animals. The task of the boom operators here is to record the sound of animals. As there are many animals and microphones, the mixing (recording) process is highly dependent on the relative positions between them. Thus, the animal (source $s_i$), loosely speaking, influences the recording (mixing function $\mathbf{f}$) through its position. The moving direction and speed of them can be loosely interpreted as the partial derivative w.r.t. the $i$-th source, i.e. $\partial \mathbf{f} / \partial s_i$. Assuming that the wild animals are not cooperative enough to fine-tune their positions and speeds for a better recording and the safari park is not crowded, the influences of animals on the recording are generated independently by them in the sense that they are not affected by the others while moving. As a result of that generating process, the column vectors of the Jacobian of the mixing function are uncorrelated with each other and the partial derivative w.r.t. to each source is independent of other sources. Other practical scenarios include fields that adopt independent influences in process like orthogonal coordinate transformations (Gresele et al., 2021), such as dynamic control (Mistry et al., 2010) and structural geology (De Paor, 1983).

Besides, similar notion of independent influences may be fundamental to tasks related to representation learning. Taking visual representation for instance, the process of generating observable images from latent semantics, e.g., drawing when thinking of specific objects or digital rendering, might be naturally a set of independent mechanisms (Rao & Ruderman, 1998). This is somehow consistent with the empirical results in previous works that flow-based generative models (Sorrenson et al., 2020) and variational autoencoders (Khemakhem et al., 2020) are able to reconstruct real-world images pretty well, even when it is not clear whether the generative process satisfies their assumptions on the auxiliary variables (e.g., sufficient variability) because the true real-world generating process is always unknown. Thus we believe that the requirement of auxiliary variables might not be necessary in some cases, especially when the true generating process consists of a set of independent mechanisms.

Having said that, a violation of the assumption of independent influences in real-world scenarios could be ascribed to a deliberate global adjustment of sources. For example, cinema audio systems are carefully adjusted to achieve a homogeneous sound effect on every audience. The position and orientation of each speaker are fine-tuned according to the others. In this case, it may be difficult to distinguish the influences from different speakers because of the fine-tuning. This may lead to a high degree of multicollinearity across the columns of Jacobian, thus violating the assumption of independent influences. Other possible violations can also be found in similar scenarios where the influences from different sources are fine-tuned in order to achieve homogeneity (e.g., the cocktail party problem described by Gresele et al. (2021)), or in an unlikely scenario where the mixing process is naturally component-wise (e.g., an autoregressive model (Papamakarios et al., 2021)). Future works include validating the proposed conditions in complicated real-world tasks.

Regarding Assumption (iii), we first note that mixing functions with factorial Jacobian determinants are of a much wider range as compared to component-wise transformations. To see this, a straightforward example is the volume-preserving transformation, which has been widely adopted in generative models like those based on flows (Zhang et al., 2021). In fact, all transformations with constant Jacobian determinant can be factorized w.r.t. the sources, which, generally speaking, are not component-wise. Besides, there exists other non-volume-preserving transformations with

factorial Jacobian determinant,[1] as demonstrated by the empirical results in Yang et al. (2022), though our current knowledge of this type of function is rather limited. Moreover, the volume-preserving assumption has been demonstrated by Sorrenson et al. (2020); Yang et al. (2022) to be helpful to weaken the requirement of auxiliary variables, specifically by reducing the number of required labels. Our work takes this further by imposing a weaker version, i.e., Assumption (iii), which, together with the other constraints, removes the need for any auxiliary variable.

Proposition 2.1 demonstrates the possibility of the identifiability of nonlinear ICA while maintaining the standard mutually marginal independence assumption of sources. This is consistent with the original setting of ICA and plays an important role in a more general range of unsupervised learning tasks, compared to previous works relying on conditional independence given auxiliary variables. In fact, one may modify Proposition 2.1 to consider the generating process based on auxiliary variables by imposing similar restrictions on the mixing process. Different from previous works (Hyvärinen & Morioka, 2016; 2017; Hyvärinen et al., 2019; Lachapelle et al., 2022; Yang et al., 2022), this extension removes common restrictions on the required auxiliary variables, such as the number of distinct values (e.g., labels), non-proportional variances, or sufficient variability. Therefore, although our main result (Proposition 2.1) focuses on nonlinear ICA with unconditional priors, it provides potential insight into improving the flexibility of utilizing the auxiliary variable when it is available. As a trade-off, the additional assumptions on the mixing process might limit its usage.

## 2.2 Identifiability up to component-wise linear transformation

Most works in nonlinear ICA focus on identifiability up to a component-wise invertible transformation and a permutation (Hyvärinen & Morioka, 2016; Hyvärinen et al., 2019; Hälvä & Hyvärinen, 2020; Khemakhem et al., 2020), which is also called *permutation-identifiability* in the literature of disentanglement (Lachapelle et al., 2022). This indeterminacy is trivial compared to the fundamental nonuniqueness of nonlinear ICA (Hyvärinen & Pajunen, 1999) and is analogous to the indeterminacy involving rescaling and permutation in linear ICA (Hyvärinen & Pajunen, 1999). Recently, Yang et al. (2022) provided an identifiability result based on auxiliary variables that further reduces the indeterminacy of the component-wise nonlinear transformation to a linear one. Inspired by it, the following result provides the identifiability conditions of nonlinear ICA up to a component-wise linear transformation with unconditional priors. This reduced indeterminacy could give rise to a more informative disentanglement and opens up the possibility for further improving the quality of recovery with only element-wise linear operations, such as the Hadamard product. The result is formally stated as follows, with a proof provided in Appendix A.2.

**Proposition 2.2.** *Besides the assumptions in Proposition 2.1, assume that the source distribution $p_{\mathbf{s}}(\mathbf{s})$ is a factorial multivariate Gaussian and function $f$ is volume-preserving. Then $\mathbf{h} := \mathbf{g} \circ \mathbf{f}$ is a composition of a component-wise linear transformation and a permutation.*

## 3 Experiments

**Setup.** Based on the proposed results, we use a volume-preserving flow to recover the sources up to the indeterminacies mentioned in Section 2.1. Similar to (Gresele et al., 2021), we consider a regularized maximum likelihood estimation to enforce orthogonality between columns of the Jacobian. Differently, we propose an objective to unify the norm of columns in the estimation, which, with a slight abuse of notation, is as follows:

$$\mathcal{L}(\mathbf{g}; \mathbf{x}) = \mathbb{E}_{\mathbf{x}} \left[ \log p_{\mathbf{g}}(\mathbf{x}) - \lambda_1 \left( n \log \left( \frac{1}{n} \sum_{i=1}^{n} \left\| \frac{\partial \mathbf{g}}{\partial x_i} \right\|_2 \right) - \log |\det(\mathbf{J}_{\mathbf{g}}(\mathbf{x}))| \right) \right],$$

where $\lambda_1$ is a regularization parameter and $\| \cdot \|_2$ denotes the Euclidean norm. The following proposition summarizes its related properties:

**Proposition 3.1.** *The following inequality holds*

$$n \log \left( \frac{1}{n} \sum_{i=1}^{n} \left\| \frac{\partial \mathbf{g}}{\partial x_i} \right\|_2 \right) - \log |\det(\mathbf{J}_{\mathbf{g}}(\mathbf{x}))| \geq 0,$$

---

[1]A toy example: the Jacobian determinant of mixing function w.r.t. $\left( x_1 = \frac{a s_1 s_2}{a + b s_2}, x_2 = \frac{b s_1}{b + a s_2} \right)$ is $a b s_1$, where $a, b \neq 0$ are some constants.

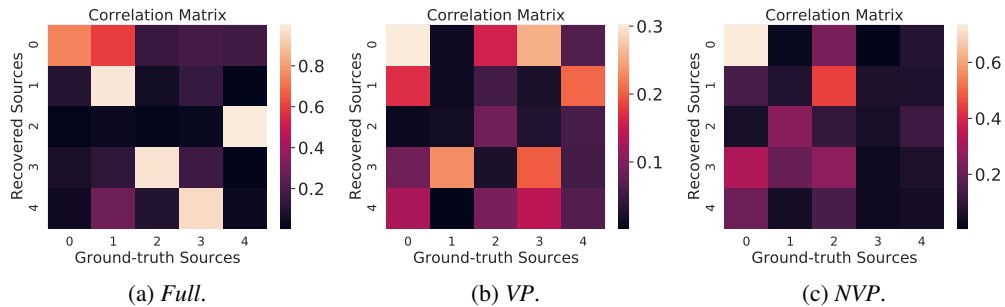

(a) *Full.*    (b) *VP.*    (c) *NVP.*

Figure 2: Pearson correlation matrices between the ground-truth and the recovered sources.

*with equality iff.* $\mathbf{J_g}(\mathbf{x}) = \mathbf{O}(\mathbf{x})\lambda(\mathbf{x})$*, where* $\mathbf{O}(\mathbf{x})$ *is an orthogonal matrix and* $\lambda(\mathbf{x})$ *is a scalar.*

The proof is provided in Appendix 3.1. We train a volume-preserving flow to maximize $\mathcal{L}(\mathbf{g}; \mathbf{x})$.

We conduct an ablation study to verify the necessity of the proposed assumptions. Specifically, we focus on the two major conditions, i.e., independent influences and factorial change of the volume. We train a volume-preserving flow to maximize the likelihood, which constrains the Jacobian determinant of GLOW (Kingma & Dhariwal, 2018). For non-volume-preserving transformations, we use the original version of GLOW instead. The observational data are generated according to the required assumptions, as detailed in Appendix B.

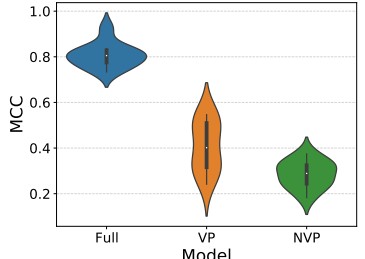

Figure 1: MCC when part of the conditions are not satisfied ($n = 5$).

**Results.** Results for each model are summarised in Figure 1. We construct each model as follows: *(Full)* Influences are mutually independent and transformations are volume-preserving; *(VP)* Compared to *Full*, the assumption of independent influences is violated; *(NVP)* Compared to *VP*, (un)mixing functions are not volume-preserving. One observes that when the proposed assumptions are fully satisfied (*Full*), our model achieves the highest mean correlation coefficient (MCC) on average. Besides, according to the decline of performance after violating each of these conditions, the necessity of them appears to be supported. The visualization of the Pearson correlation matrices is shown in Figure 2.

# 4 CONCLUSION

We provide an identifiability result for nonlinear ICA with unconditional priors. In particular, we prove that the i.i.d. latent sources, under certain assumptions, can be recovered up to a component-wise invertible transformation and a permutation with only conditions on the nonlinear mixing process. Therefore it may stay closer to the original notion of ICA that is based on the marginal independence assumption of latent sources, while previous works rely on conditional independence on auxiliary variables as weak supervision or inductive bias. Besides, by further assuming the Gaussianity of sources, we prove that the indeterminacy of the component-wise transformation can be reduced to a component-wise linear transformation, which sheds light on potential further optimizations by element-wise linear operations. By including the auxiliary variable but without common constraints on it, our results may provide flexibility in specific scenarios when auxiliary variables are available.

**Limitation and discussion.** Although our identifiability result removes the need for auxiliary variables, the assumptions on the mixing process may restrict its usage in specific scenarios as discussed in Section 2.1. This is inevitably a trade-off between introducing auxiliary variables and imposing restrictions on the mixing process in order to achieve the identifiability, whose practical usage depends on the scenario and information available. It is worth noting that Gresele et al. (2021) assumed orthogonality, conformal map and the others in order to rule out two specific types of spurious solutions (i.e., Darmois construction (Darmois, 1951) and "rotated-Gaussian" MPAs (Locatello et al., 2019)) without introducing auxiliary variables. Our future work includes validating the proposed conditions in complicated real-world tasks and generalizing some of them.

ACKNOWLEDGEMENTS

We thank the anonymous reviewers for their constructive comments and helpful criticisms. This work was supported in part by the National Institutes of Health (NIH) under Contract R01HL159805, by the NSF-Convergence Accelerator Track-D award #2134901, by the United States Air Force under Contract No. FA8650-17-C7715, and by a grant from Apple. The NIH or NSF is not responsible for the views reported in this paper.

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

## A PROOFS

### A.1 PROOF OF PROPOSITION 2.1

We restate the setup here. Consider the following data-generating process of nonlinear ICA

$$p_{\mathbf{s}}(\mathbf{s}) = \prod_{i=1}^{n} p_{s_i}(s_i),$$

$$\mathbf{x} = \mathbf{f}(\mathbf{s}),$$

where $\mathbf{x}$ denotes the observed random vector, $\mathbf{s}$ is the latent random vector representing the marginally independent sources, $p_{s_i}$ is the marginal probability density function (pdf) of $s_i$, $p_{\mathbf{s}}$ is the joint pdf, and $\mathbf{f} : \mathbf{s} \to \mathbf{x}$ denotes a nonlinear mixing function. We preprocess the observational data by centering the columns of the Jacobian of the mixing function. This could be achieved by applying a fixed transformation on the observational data that is equivalent to left multiplying the Jacobian of the mixing function with the centering matrix $\mathbf{U} := \mathbf{I}_n - (1/n)\mathbf{1}\mathbf{1}^\top$, where $\mathbf{I}_n$ is the identity matrix of size $n$ and $\mathbf{1}$ is the $n$-dimensional vector of all ones. After centering, according to the assumption of independent influences, the columns of the Jacobian matrix are uncorrelated and of zero-means, and thus orthogonal to each other [2]. Since the preprocessing transformation $\mathbf{U}$ is fixed w.r.t. no variables other than $n$, which is the fixed number of dimensions, it could always be recovered after reconstruction.

We assume that the distribution of the estimated sources $\hat{\mathbf{s}}$ follows a factorial multivariate Gaussian:

$$p_{\hat{\mathbf{s}}}(\hat{\mathbf{s}}) = \prod_{i=1}^{n} \frac{1}{Z_i} \exp\left(-\theta_{i,1}\hat{s}_i - \theta_{i,2}\hat{s}_i^2\right),$$

where $Z_i > 0$ is a constant. The sufficient statistics $\theta_{i,1} = -\frac{\mu_i}{\sigma_i^2}$ and $\theta_{i,2} = \frac{1}{2\sigma_i^2}$ are assumed to be linearly independent. We set the variances $\sigma_i^2$ to be distinct without loss of generality, which can be placed as a constraint during the estimation process as the sufficient statistics of the estimated distribution are trainable.

Our goal here is to show that function $\mathbf{h} = \mathbf{g} \circ \mathbf{f}$ is a composition of a permutation and a component-wise invertible transformation of sources. By using chain rule repeatedly, we write the Jacobian of function $\mathbf{h} := \mathbf{g} \circ \mathbf{f}$ as

$$\begin{aligned} \mathbf{J_h}(\mathbf{s}) &= \mathbf{J_{g \circ f}}(\mathbf{s}) \\ &= \mathbf{J_g}(\mathbf{f}(\mathbf{s}))\mathbf{J_f}(\mathbf{s}) \\ &= \mathbf{J_g}(\mathbf{x})\mathbf{J_f}(\mathbf{s}) \\ &= \hat{\mathbf{O}}(\mathbf{x})\hat{\lambda}(\mathbf{x})\mathbf{O}(\mathbf{s})\mathbf{D}(\mathbf{s}) \\ &= \hat{\lambda}(\mathbf{x})\hat{\mathbf{O}}(\mathbf{x})\mathbf{O}(\mathbf{s})\mathbf{D}(\mathbf{s}), \end{aligned} \tag{4}$$

where $\hat{\lambda}(\mathbf{x})$ is a scalar, $\hat{\mathbf{O}}(\mathbf{x})$ and $\mathbf{O}(\mathbf{s})$ represent the corresponding orthogonal matrices, and $\mathbf{D}(\mathbf{s})$ denotes the corresponding diagonal matrix. The fourth equality follows from assumption i.

Since $\mathbf{J_h}(\mathbf{s}) = \hat{\lambda}(\mathbf{x})\hat{\mathbf{O}}(\mathbf{x})\mathbf{O}(\mathbf{s})\mathbf{D}(\mathbf{s})$ by Eq. 4 and $\hat{\mathbf{O}}(\mathbf{x})\mathbf{O}(\mathbf{s})$ is an orthogonal matrix, the columns of $\mathbf{J_h}(\mathbf{s})$ are orthogonal to each other, i.e.,

$$\mathbf{J_h}(\mathbf{s})_{:,j}^\top \mathbf{J_h}(\mathbf{s})_{:,k} = 0, \quad \forall j \neq k,$$

where $\mathbf{J_h}(\mathbf{s})_{:,j}$ is the $j$-th column of matrix $\mathbf{J_h}(\mathbf{s})$. This can be rewritten as

$$\sum_{i=1}^{n} \left(\frac{\partial h_i}{\partial s_j}\frac{\partial h_i}{\partial s_k}\right) = 0, \quad \forall j \neq k. \tag{5}$$

---

[2]Similarly, Gresele et al. (2021) directly formalize the notion of Independent Mechanism Analysis as orthogonality of the columns of the Jacobian.

By assumption, the components of the true sources $\mathbf{s}$ are mutually independent, while the estimated sources $\hat{\mathbf{s}}$ follows a multivariate Gaussian distribution:

$$p_{\mathbf{s}}(\mathbf{s}) = \prod_{i=1}^n p_{\mathbf{s}_i}(s_i),$$

$$p_{\hat{\mathbf{s}}}(\hat{\mathbf{s}}) = \prod_{i=1}^n \frac{1}{Z_i} \exp\left(-\theta_{i,1}\hat{s}_i - \theta_{i,2}\hat{s}_i^2\right), \tag{6}$$

where $Z_i > 0$ is a constant. The sufficient statistics $\theta_{i,1}$ and $\theta_{i,2}$ are assumed to be linearly independent.

Applying the change of variable rule, we have $p_{\mathbf{s}}(\mathbf{s}) = p_{\hat{\mathbf{s}}}(\hat{\mathbf{s}})|\det(\mathbf{J_h}(\mathbf{s}))|$, which, by plugging in Eq. 6 and taking the logarithm on both sides, yields

$$\sum_{i=1}^n \log p_{s_i}(s_i) - \log|\det(\mathbf{J_h}(\mathbf{s}))| = -\sum_{i=1}^n \left(\theta_{i,1}h_i(\mathbf{s}) + \theta_{i,2}h_i(\mathbf{s})^2 + \log Z_i\right). \tag{7}$$

According to assumption iii, we have the factorization $\det(\mathbf{J_h}(\mathbf{s})) = \prod_{i=1}^n y_i(s_i)$, where $y_i$ is a function that depends only on $s_i$. This implies that

$$\sum_{i=1}^n \left(\log p_{s_i}(s_i) - |y_i(s_i)|\right) = -\sum_{i=1}^n \left(\theta_{i,1}h_i(\mathbf{s}) + \theta_{i,2}h_i(\mathbf{s})^2 + \log Z_i\right). \tag{8}$$

Taking second-order derivatives of both sides of Eq. 8 w.r.t. $s_j$ and $s_k$ with $j \neq k$, we obtain

$$\sum_{i=1}^n \left((\theta_{i,1} + 2\theta_{i,2}h_i(\mathbf{s}))\frac{\partial^2 h_i}{\partial s_j \partial s_k} + 2\theta_{i,2}\frac{\partial h_i}{\partial s_j}\frac{\partial h_i}{\partial s_k}\right) = 0, \quad \forall j \neq k. \tag{9}$$

According to the assumption of independent influences as well as corresponding restriction on the estimating process, for all $j \neq k$, we have $\frac{\partial^2 h_i}{\partial s_j \partial s_k} = 0$. Since $\theta_{i,1}$ and $\theta_{i,2}$ are sufficient statistics of a multivariate Gaussian, we have $\theta_{i,2} = \frac{1}{2\sigma_i^2}$, where $\sigma_i$ represents the standard deviation. Plugging it into Eq. 9 yields

$$\sum_{i=1}^n \left(\frac{1}{\sigma_i^2}\frac{\partial h_i}{\partial s_j}\frac{\partial h_i}{\partial s_k}\right) = 0, \quad \forall j \neq k. \tag{10}$$

Note that a similar formulation of Eq. 10 is also derived and presented in Lemma 1 by Yang et al. (2022). The only difference is the term $y_i(s_i)$, which does not affect the formulation as it has no cross terms. Thus, based on Lemma 1 in (Yang et al., 2022), we can also obtain Eq. 10 with the current assumptions or alternatively weaker ones.

Then, we define the following matrices:

$$\mathbf{\Lambda}_1 = \mathrm{diag}\left(\frac{1}{\sigma_1^2}, \cdots, \frac{1}{\sigma_n^2}\right),$$

$$\mathbf{\Lambda}_2 = \mathbf{I}_n,$$

$$\mathbf{\Sigma}(1, \mathbf{s}) = \mathrm{diag}\left(\sum_{i=1}^n \frac{1}{\sigma_i^2}\left(\frac{\partial h_i}{\partial s_1}\right)^2, \cdots, \sum_{i=1}^n \frac{1}{\sigma_i^2}\left(\frac{\partial h_i}{\partial s_n}\right)^2\right),$$

$$\mathbf{\Sigma}(2, \mathbf{s}) = \mathrm{diag}\left(\sum_{i=1}^n \left(\frac{\partial h_i}{\partial s_1}\right)^2, \cdots, \sum_{i=1}^n \left(\frac{\partial h_i}{\partial s_n}\right)^2\right). \tag{11}$$

We rewrite Eq. 10 and Eq. 5 as

$$\begin{cases} \mathbf{J_h}(\mathbf{s})^\top \mathbf{\Lambda}_1 \mathbf{J_h}(\mathbf{s}) = \mathbf{\Sigma}(1, \mathbf{s}), \\ \mathbf{J_h}(\mathbf{s})^\top \mathbf{\Lambda}_2 \mathbf{J_h}(\mathbf{s}) = \mathbf{\Sigma}(2, \mathbf{s}). \end{cases} \tag{12}$$

Because all elements in matrices defined in Eq. 11 are positive, we can take sqaure roots on both sides of equations in Eq. 12, yielding

$$
\begin{cases}
\mathbf{\Lambda}_1^{1/2}\mathbf{J_h}(\mathbf{s}) = \mathbf{O}_1\mathbf{\Sigma}(1,\mathbf{s})^{1/2}, \\
\mathbf{\Lambda}_2^{1/2}\mathbf{J_h}(\mathbf{s}) = \mathbf{O}_2\mathbf{\Sigma}(2,\mathbf{s})^{1/2},
\end{cases}
\tag{13}
$$

where $\mathbf{O}_1$ and $\mathbf{O}_2$ are the corresponding orthogonal matrices ($\mathbf{O}_1^\mathsf{T}\mathbf{O}_1 = \mathbf{O}_2^\mathsf{T}\mathbf{O}_2 = \mathbf{I}_n$) and $\mathbf{\Lambda}^{1/2}$ denotes the matrix by applying entry-wise square root to the entries of diagonal matrix $\mathbf{\Lambda}$. Then we rewrite Eq. 13 as

$$
\begin{cases}
\mathbf{J_h}(\mathbf{s}) = \mathbf{\Lambda}_1^{-1/2}\mathbf{O}_1\mathbf{\Sigma}(1,\mathbf{s})^{1/2}, \\
\mathbf{J_h}(\mathbf{s}) = \mathbf{\Lambda}_2^{-1/2}\mathbf{O}_2\mathbf{\Sigma}(2,\mathbf{s})^{1/2}.
\end{cases}
$$

By reformulating, we have

$$
\mathbf{\Sigma}(1,\mathbf{s})^{1/2}\mathbf{\Sigma}(2,\mathbf{s})^{-1/2} = \mathbf{O}_1^{-1}\mathbf{\Lambda}_1^{1/2}\mathbf{\Lambda}_2^{-1/2}\mathbf{O}_2.
\tag{14}
$$

We consider both sides of Eq. 14 as a singular value decomposition of the LHS. Since $\sigma_j \neq \sigma_k, \forall j \neq k$, elements in $\mathbf{\Lambda}_1^{1/2}\mathbf{\Lambda}_2^{-1/2}$ are distinct. Therefore, non-zero elements in $\mathbf{\Lambda}_1^{1/2}\mathbf{\Lambda}_2^{-1/2}$ are also distinct because $\mathbf{\Lambda}_2$ is an identity matrix.

Because the mixing function is assumed to be invertible, the Jacobian of the mixing function $\mathbf{J_f}(\mathbf{s}) = \mathbf{O}(\mathbf{s})\mathbf{D}(\mathbf{s})$ is of full-rank. Therefore, all diagonal elements in the diagonal matrix $\mathbf{\Lambda}_1^{1/2}\mathbf{\Lambda}_2^{-1/2}$ are non-zero.

Then, based on the uniqueness of singular value decomposition (see e.g. Proposition 4.1 in Trefethen & Bau III (1997)), $\mathbf{O}_2$ is a composition of a permutation matrix and a signature matrix. Therefore, $\mathbf{J_h}(\mathbf{s}) = \mathbf{\Lambda}_2^{-1/2}\mathbf{O}_2\mathbf{\Sigma}(2,\mathbf{s})^{1/2}$ is a generalized permutation matrix. It thus follows that the function $\mathbf{h} := \mathbf{g} \circ \mathbf{f}$ is a composition of a component-wise invertible transformation and a permutation.

## A.2 Proof of Proposition 2.2

**Proposition 2.2.** *Besides the assumptions in Proposition 2.1, assume that the source distribution $p_\mathbf{s}(\mathbf{s})$ is a factorial multivariate Gaussian and function $f$ is volume-preserving. Then $\mathbf{h} := \mathbf{g} \circ \mathbf{f}$ is a composition of a component-wise linear transformation and a permutation.*

*Proof.* Since $\mathbf{s}$ is from a factorial multivariate Gaussian distribution, we represent its density as

$$
p_\mathbf{s}(\mathbf{s}) = \prod_{i=1}^n \frac{1}{Z_i} \exp\left(-\theta'_{i,1}s_i - \theta'_{i,2}s_i^2\right).
$$

According to Proposition 2.1, function $\mathbf{h}$ is a component-wise invertible transformation of sources, i.e.,

$$
h_i(\mathbf{s}) = h_i(s_i).
$$

Combining the above two equations with Eq. 7 and the condition that $\mathbf{h}$ is volume-preserving, it follows that

$$
\sum_{i=1}^n \left(-\theta'_{i,1}s_i - \theta'_{i,2}s_i^2\right) = -\sum_{i=1}^n \left(\theta_{i,1}h_i(s_i) + \theta_{i,2}h_i(s_i)^2 + \log Z_i\right).
$$

Then we have

$$
\theta'_{i,1}s_i + \theta'_{i,2}s_i^2 + \log Z_i = \theta_{i,1}h_i(s_i) + \theta_{i,2}h_i(s_i)^2.
$$

Therefore, $h_i(s_i)$ is a linear function of $s_i$. $\square$

## A.3 Proof of Proposition 3.1

**Proposition 3.1.** *The following inequality holds*

$$
n \log\left(\frac{1}{n}\sum_{i=1}^n \left\|\frac{\partial \mathbf{g}}{\partial x_i}\right\|_2\right) - \log|\det(\mathbf{J_g}(\mathbf{x}))| \geq 0,
$$

*with equality iff. $\mathbf{J_g}(\mathbf{x}) = \mathbf{O}(\mathbf{x})\lambda(\mathbf{x})$, where $\mathbf{O}(\mathbf{x})$ is an orthogonal matrix and $\lambda(\mathbf{x})$ is a scalar.*

*Proof.* According to Hadamard's inequality, we have

$$\sum_{i=1}^{n} \log \left\| \frac{\partial \mathbf{g}}{\partial x_i} \right\|_2 - \log |\det(\mathbf{J_g}(\mathbf{x}))| \geq 0,$$

with equality iff. vectors $\frac{\partial \mathbf{g}}{\partial x_i}, i = 1, 2, \ldots, n$ are orthogonal. Then applying the inequality of arithmetic and geometric means yields

$$n \log \left( \frac{1}{n} \sum_{i=1}^{n} \left\| \frac{\partial \mathbf{g}}{\partial x_i} \right\|_2 \right) - \log |\det(\mathbf{J_g}(\mathbf{x}))| \geq \sum_{i=1}^{n} \log \left\| \frac{\partial \mathbf{g}}{\partial x_i} \right\|_2 - \log |\det(\mathbf{J_g}(\mathbf{x}))| \geq 0,$$

with the first equality iff. for all $i = 1, 2, \ldots, n$, $\left\| \frac{\partial \mathbf{g}}{\partial x_i} \right\|_2$ is equal. Because $\mathbf{J_g}(\mathbf{x})$ is non-singular, $\frac{\partial \mathbf{g}}{\partial x_i}$ are orthogonal and equal to each other for all $i = 1, 2, \ldots, n$ iff. $\mathbf{J_g}(\mathbf{x}) = \mathbf{O}(\mathbf{x})\lambda(\mathbf{x})$. □

## B  EXPERIMENTS

### B.1  SYNTHETIC DATASETS AND IMPLEMENTATION DETAILS

In order to generate observational data satisfying required assumptions, we simulate the sources and mixing process as follows:

***Full***.    Following Gresele et al. (2021), we generate the mixing functions based on Möbius transformations with scaled sources and constant volume. According to Liouville's results (Flanders, 1966) (also summarized in Theorem F.2 in Gresele et al. (2021)), the Möbius transformation guarantees the orthogonality between the columns vectors of its Jacobian matrix, which achieves uncorrelatedness after centering. Different from Gresele et al. (2021), we scaled the sources while preserving volumes before Möbius transformation with distinct scalers to make sure that the generating process is not a conformal map.

***VP***.    Here we describe the generating process that is volume-preserving but not necessarily with orthogonal columns of Jacobian. We use volume-preserving flow called GIN (Sorrenson et al., 2020) to generate the mixing function. GIN is a volume-preserving version of RealNVP (Dinh et al., 2016), which achieves volume preservation by setting the scaling function of the final component to the negative sum of previous ones.[3] We use the official implementation of GIN (Sorrenson et al., 2020), which is part of FrEIA.

***NVP***.    Here we describe the generating process without assumptions on volume preservation and orthogonal columns of Jacobian. Following (Sorrenson et al., 2020), we use GLOW (Kingma & Dhariwal, 2018) to generate the mixing function. The difference between the coupling block in GLOW and GIN is that the Jacobian determinant of the former is not constrained to be one. The implementation of GLOW is also included in the official implementation of GIN (Sorrenson et al., 2020), which is also part of FrEIA.

Regarding the model evaluation, we use the mean correlation coefficient (MCC) between the ground-truth and recovered latent sources. We first compute pair-wise correlation coefficients between the true sources and recovered ones. Then we solve a linear sum assignment problem to match each recovered source to the ground truth with the highest correlation between them. MCC is a standard metric to measure the degree of identifiability up to component-wise transformation in the literature (Hyvärinen & Morioka, 2016).

Following previous works (Khemakhem et al., 2020), the ground-truth sources ($n = 5$) are sampled from a multivariate Gaussian, with zero means and variances sampled from a uniform distribution on $[0.5, 3]$[4]. The sample size for all experiments is 10000. The learning rate is 0.01 and batch size is 1000. The number of coupling layers for both GIN and GLOW is set as 24. The estimating model is built in the reverse direction as described above. The code of our experiments is in Python and developed based on GIN (Sorrenson et al., 2020).

---

[3]In the official implementation of GIN, the volume-preservation is achieved in a slightly different way compared to that in its original paper for better stability of training, but there is no difference in the theoretical result w.r.t. volume-preservation.

[4]These are of the same values as previous works (Khemakhem et al., 2020; Sorrenson et al., 2020).

