# OpenReview forum: "On the Identifiability of Nonlinear ICA with Unconditional Priors"
_ICLR.cc/2022/Workshop/OSC — ICLR2022 OSC  Oral_

### Official Review · Reviewer_HwLf · 2022-03-15
**Nonlinear ICA**

**Rating:** 3
**Confidence:** 3

**Review:**

This paper solves a long standing open problem, namely the identifiability of nonlinear independent component analysis. It shows that nonlinear ICA is identifiable under three fairly natural and general assumptions. The assumptions do not include auxiliary variables, as used in some recent related work, and thus stays closer to the original notion of ICA. Some small scale experiments show that nonlinear ICA is indeed possible in practice, though much work remains to be done to demonstrate this convincingly.

I checked the proof and the idea makes sense to me and it seems fairly rigorous, though I did not spend enough time with it to become 100% certain of the correctness.

In my opinion this is a very significant result. The only question I have, which can be answered in followup work, is how general the class of functions is that has factorizable Jacobian determinant / how reasonable this assumption is. Part of the answer could also be empirical; show that the method works on hard real-world problems.

---

### Official Review · Reviewer_Kdis · 2022-03-16
**On the Identifiability of Nonlinear ICA with Unconditional Priors**

**Rating:** 3
**Confidence:** 2

**Review:**

Overall: Paper is very clear and contribution is sound. I would implore the authors to ensure it is clear to the reader how (Yang et al., 2022) served as a foundation for this result both at a high-level in the main text, and at a low-level with details in the appendix. Furthermore, the contribution stands alone, but it would be beneficial to push for a real-world analysis of the assumptions to be done in future work.

Specific points:
- " This implicitly restricts p_{s}(s)". How? It is important to be precise about how your assumptions on the estimated model restrict the ground-truth data generating process. As a rule, the data generating process is separate, the estimated model should not affect it. Do you instead mean that you must restrict the data generating process to prove identifiability for the given estimated model? Please clarify.
- Footnote 3 on pg. 9 should be included in the assumptions.
- It should be justified why setting $\theta_{i,1}=0$ can be done w.l.o.g.
- Not immediately clear how $O_1$ and $O_2$ are introduced in eq. (17).
- Should be fully clarified how Thm. 2.1 differs from any of the theoretical results in (Yang et al., 2022). It's entirely fine that it was used as a resource is proving the statement, but it would be to the reader's credit to understand what has been proven here that wasn't proven in (Yang et al., 2022)? While the following statement does clarify things ("Moreover, the volume-preserving assumption...completely removes the need for any auxiliary variable."), a more precise theoretical comparison to (Yang et al., 2022) would be welcome. Clarifying the innovations upon (Yang et al., 2022) the authors performed to be able to "completely remove the need for any auxiliary variable" would aid readers in using said innovations in future progress. Furthermore, it would be beneficial to clarify what "labels" is referring to when used in the aforementioned statement, and altogether how it is used in the paper, "number of distinct values" is still vague, what values?
- While the intuition building paragraphs were helpful, it is my opinion that they will be better served in the introduction. It should be said, in fear of stating the obvious, that if the authors would like to justify their assumptions as reasonable, is to rigorously analyze real-world data to see whether the assumptions are or aren't satisfied, and if a model satisfies the given assumption, whether the identifiability criterion is satisfied. While this is hard due to the data generating process for real-world data being unknown, works exist [1] which, at the very least, have made an attempt at justification. If this is possible, this would be significantly more convincing, in my view, than the arguments from intuition currently presented.
- " As a trade-off, the additional assumptions on the mixing process might limit its usage in some scenarios." Not to harp on this point, but a real-world analysis on whether the additional assumptions on the mixing process does limit usage in some scenarios, particularly learning in which scenarios is and isn't it limiting, should at least be suggested for future work.
- Once again, what is the additional novelty/innovation, if any, in Theorem 2.2 relative to (Yang et al., 2022), besides what it inherits from Theorem 2.1, should be clarified for the reader.
- Minor grammatical errors should be checked, i.e. " columns of Jacobian of estimating function" -> "columns of the Jacobian of the estimating function".

---

### Decision · Program_Chairs · 2022-03-19

**Decision:**

Accept (Oral)

**Comment:**

Both reviewers agree about the high quality of the paper and the significance of the results. Reviewer HwLf highlights that this paper provides a solution to the long-standing problem of identifiability of nonlinear ICA, which -- together with the high quality of the submission -- more than justifies an oral presentation at the workshop.

The authors are encouraged to take the extensive feedback by the reviewers into account when preparing the camera-ready version of the paper.